# Unpredictable Repeated Stress in Rainbow Trout (*Oncorhynchus mykiss*) Shifted the Immune Response against a Fish Parasite

**DOI:** 10.3390/biology13100769

**Published:** 2024-09-27

**Authors:** Cyril Henard, Hanxi Li, Barbara F. Nowak, Louise von Gersdorff Jørgensen

**Affiliations:** 1Department of Veterinary and Animal Sciences, University of Copenhagen, Stigbøjlen 7 Frederiksberg C, 1870 Frederiksberg, Denmark; lvgj@sund.ku.dk; 2Department of Health Technology, Technical University of Denmark, 2800 Kongens Lyngby, Denmark; hanxli@dtu.dk; 3Institute for Marine and Antarctic Studies, University of Tasmania, Launceston 7248, Tasmania, Australia; b.nowak@utas.edu.au

**Keywords:** fish handling, immune gene expression, immune signaling, parasite infection, *Ichthyophthirius multifiliis*

## Abstract

**Simple Summary:**

The aquaculture industry is a major contributor to the supply of animal proteins. Parasitic diseases like the white sport disease caused by *Ichthyophthirius multifiliis* (ich) is a significant threat to fish production and welfare. Previous studies explored the fish immune response against ich. However, the effects of farming procedures such as fish handling were not studied in the context of white spot disease. Our studies investigated the impact of handling procedures (air exposure, chasing, transfer) on the fish’s ability to fight the parasite compared to non-handled fish. The results indicated that the stress caused by handling did not change the amount of parasite detected in the fish, in the tank’s water, or the fish’s mortality. Nonetheless, several parameters of the fish immune response were significantly altered due to the handling procedures. Overall, our results suggested that the handling procedures investigated did not impair the fish’s ability to fight the parasite.

**Abstract:**

Farmed fish are regularly subjected to various stressors due to farming practices, and their effect in the context of a disease outbreak is uncertain. This research evaluated the effects of unpredictable repeated stress in rainbow trout challenged with the ciliate *Ichthyophthirius multifiliis*, known to cause white spot disease in freshwater fish. Before and after the pathogen exposure, fish were handled with a random rotation of three procedures. At 7 days post-infection (dpi), the parasite burden was evaluated in fish and in the tank’s water, and the local and systemic immune responses were investigated in the gill and spleen, respectively. The fish mortality was recorded until 12 dpi, when all the fish from the infected groups died. There was no statistical difference in parasite burden (fish and tank’s water) and infection severity between the two infected fish groups. The immune gene expression analysis suggested a differential immune response between the gill and the spleen. In gills, a T helper cell type 2 immune response was initiated, whereas in spleen, a T helper cell type 1 immune response was observed. The stress has induced mainly upregulations of immune genes in the gill (*cat-1*, *hep*, *il-10*) and downregulations in the spleen (*il-2*, *il-4*/*13a*, *il-8*). Our results suggested that the unpredictable repeated stress protocol employed did not impair the fish immune system.

## 1. Introduction

Among the parasitic diseases affecting teleost fish worldwide, ichthyophthiriosis caused by the ciliate *Ichthyophthirius multifiliis* (Ich) Fouquet (1876) may be the first ever reported (1126 BC in China), one of the most widespread and the most economically relevant affecting ornamental, farmed, and wild fish [1]. The lifecycle of Ich does not include intermediate hosts and is characterized by three distinct stages consisting of a host feeding trophont, a vegetative tomont and an infectious free swimming theront [2]. The Latin species name *multifiliis* comes from the prefix “*multi*”, meaning many, and “*filiis*”, meaning sons. This designation reflects the ability of the Ich tomont to generate many theronts, explaining partly why ichthyophthiriosis represents a major challenge if not treated promptly [3]. The Ich theronts rely on positive phototaxis for orientation; nevertheless, at close range, the free swimming stage of Ich is able to sense fish hosts (for example, through sensing their mucus or sera), which makes this disease even more dramatic in aquaria compared to the natural environment [4,5]. Furthermore, this statement is particularly valid for fish farms where fish density is higher than in the wild and fish movements are confined within a pond. In Denmark, surveys performed in rainbow trout (*Oncorhynchus mykiss*) fish farms indicated that 20 °C was the optimal temperature for ichtyophtiriosis and that these outbreaks were associated with the greatest fish mortality compared to other fish parasites [6,7].

In fish farming, the welfare of the fish is a key component to preserving the fish’s optimal immune response. The stressors perceived by the fish could have various natures (e.g., chemical, physical, social, biological) [8]. The stress responses could be designated as eustress (i.e., good stress) or distress (i.e., bad stress) that depends on the amplitude and the frequency of stress exposure [9]. In farming conditions, the fish may experience different types of stressors daily, in an unpredictable manner, meaning that stressors may occur at different times from day to day. Previous research outlined that chronic stress mitigates innate immune response and enhances T helper cell type 2 (Th2) response, whereas acute stress boosts innate immune response and T helper cell type 1 (Th1) response [10]. In the present study, we explored the effects of unpredictable repeated stress (URS) on the rainbow trout immune response against *Ichthyophthirius multifiliis*. The stressors (i.e., chasing, air exposure, and transfer) were selected for their relevance to the farming context. The impact of URS was evaluated with immune gene expression analysis. Specific molecular markers involved in inflammation (e.g., *ifn*-γ, *il-1*β, *il-10*), antimicrobial activity (e.g., *hep*, *cat-1*, *c3*), Th1 (e.g., *il-2*), and Th2 (e.g., *il-4*/*13a*) responses were examined.

## 2. Material and Method

### 2.1. Fish

The specific-pathogen-free rainbow trout used in this study were provided by the Bornholm’s Salmon Hatchery (Nexø, Denmark). This hatchery is certified free from bacterial and viral fish pathogens, according to the Danish Food Agency. The rainbow trout (12.88 g ± 2.83) were allocated to eight 80 L tanks (filled with 60 L of tap water) with 30 fish per tank and acclimated for forty-four days before the experiment started. At that stage of development, both innate and adaptative immune systems are functional in rainbow trout [11]. The water temperature was 18 °C, 20 L of water were changed daily, and the light/dark cycle was 14/10 h. The fish were fed daily with dry manufactured feed (INICIO Plus 1.1 mm, BioMar, Aarhus, Denmark) at 8 a.m.

### 2.2. Parasite

The ciliate *Ichthyophthirius multifiliis* was collected from a Danish rainbow trout fish farm located in Jutland (Denmark). Rainbow trout from the fish farm (*n* = 6) were transported in an oxygenated container to the experimental facilities at the University of Copenhagen. The rainbow trout were placed in tanks with goldfish (*Carassius auratus*) from the pet shop. The goldfish was used as a host because they are more resistant to Ich infection, and consequently, the amount of tomonts collected from one fish was superior compared to rainbow trout. The trophonts, or feeding stage, are located in the fins and the skin. After the death of the host, trophonts escape the host and become free swimming tomonts. A heavily infected goldfish fin was incubated for 4 h at 18 °C in a petri dish to collect tomonts. After incubation of tomonts for 48 h at 18 °C, theronts were released from tomonts and resuspended in 200 mL facility water (filtered 0.2 µm cellulose acetate, ADVANTEC^®^, Tokyo, Japan) and counted (10 replicates of 5 µL) under a dissecting microscope (Leica MZ9.5, Brønshøj, Denmark) to estimate the theronts concentration.

### 2.3. Unpredictable Repeated Stress, Challenge, and Sampling

The four experimental treatments were performed in duplicate (*n* = 30 fish/tank; *n* = 60 fish/treatment) tanks and defined as: control, control-stress, infected, infected-stress. Three days before the challenge, the fish from the stress groups (control-stress and infected-stress) were stressed daily once a day until the challenge, and then twice a day starting the day after theronts exposure until 8 days post infection (dpi) (Table 1). At 9 and 10 dpi, the fish were stressed once a day because of disease progression, and at 11 dpi, the stress treatment was interrupted as the reinfection of the fish by the theronts due to the second cycle of Ich caused additional stress in fish. Three types of stressors, inspired by Piato et al. (2011), were used in rotation and included air exposure (contact with air for 20 s), chasing (chased with a net for 2 min), and transfer (transfer into a 20 L tank for 2 min followed by return to their respective tank) [12]. The fish were stressed within two time frames (9:00–11:00 a.m. and 1:00–4:00 pm), and therefore, our protocol was qualified as unpredictable repeated stress. The two time frames were selected to provide enough time for the fish to recover from the handling procedure. On the day of infection, 500 theronts per fish (i.e., 60,000 theronts per tank) were introduced to the two infection group tanks (infected and infected-stress) to induce a mild infection, and the pumps in the tanks were switched off for 4 h. At 7 dpi, five were sampled in each tank. The fish were euthanized with an overdose (400 mg/L) of MS-222 (tricaine methane sulphonate, Sigma-Aldrich, Brøndby, Denmark). For each sampled fish, trophonts (i.e., white spots) were counted on the left side of the fish (skin and gill arches) in the anterior posterior axis, and the fish left gill and the spleen were sampled and put in RNAlater^®^ (Sigma-Aldrich, Brøndby, Denmark) at 4 °C for 24 h, then stored at −20 °C until tissue RNA purification. The fish mortalities were recorded every second hour from when the first mortality occurred, and until all infected fish died. Fish were euthanized before their deaths when severe disease occurred. Fish euthanasia was counted as mortality, and no further analysis was performed on those fish.

### 2.4. Water Environmental DNA (eDNA) I. multifiliis Standard Curve

The theront suspension was used for the fish challenge and for the construction of a standard curve for the Ich eDNA analysis. The theront suspension for the standard curve was adjusted to 500 theronts mL^−1^ and 200 mL of the suspension was filtered on a 0.45 µm membrane (cellulose acetate, ADVANTEC^®^, Tokyo, Japan) in quadruplicates. The filters were stored dry at −20 °C until eDNA purification.

### 2.5. Water eDNA Sampling and Purification

Water samples (1 L per tank) were collected at 7, 9, and 12 dpi and stored at 4 °C until filtration. For each fish treatment and time point, two samples were collected, and therefore, no statistical analysis was performed. The samples were filtered through a 0.45 µm membrane (cellulose acetate, ADVANTEC^®^, Tokyo, Japan) until all the samples had been processed. Directly following filtration, the filters were stored dry at −20 °C until eDNA purification. The eDNA samples were purified with the DNeasy^®^ PowerWater^®^ Kit (Qiagen, Hvidovre, Denmark), following the manufacturer’s instructions. The purified samples were stored at −20 °C until qPCR analysis. To establish the standard curve, eight serial dilutions (1:10 in water) were used with UltraPure DNase/RNase-Free Distilled Water (Invitrogen, Kamstrup, Denmark).

### 2.6. Fish Samples RNA Purification

Total RNA from rainbow trout gill and spleen was purified with the GenElute™ Mammalian Total RNA Miniprep Kit (Sigma-Aldrich, Brøndby, Denmark). Gill and spleen samples were lysed with a Tissue Lyzer II (Qiagen, Hvidovre, Denmark) and a 5 mm stainless steel bead (Qiagen, Hvidovre, Denmark) for two minutes at 20 hz. RNA purification was performed following the manufacturer’s instructions. After RNA purification, samples were treated to remove DNA as follows: per sample, 2.5 µL of DNase I Amplification Grade (Sigma-Aldrich, Brøndby, Denmark) and 2.5 µL of 10× Reaction Buffer (Sigma-Aldrich, Brøndby, Denmark) were added, and the samples were incubated at 37 °C for 30 min for DNase treatment. To stop the DNase treatment, 2.5 µL of Stop solution (Sigma-Aldrich, Brøndby, Denmark) was added, and the samples were incubated at 65 °C for 10 min. RNA concentration was assessed with NanoDrop 2000 (Thermo Scientific, Lillerøde, Denmark). RNA integrity was assessed with electrophoresis using agarose gel (1.5%) containing ethidium bromide. The samples were loaded with RNA Sample Loading Buffer (Sigma-Aldrich, Brøndby, Denmark). The purified RNA samples were stored at −80 °C until cDNA synthesis.

### 2.7. Fish Samples cDNA Synthesis

The cDNA synthesis was performed with the TaqMan^®^ Reverse Transcription Kit (Applied Biosystems, Lillerøde, Denmark). The reaction volume per sample was composed of 10 µL of mastermix (1.16 µL H_2_0, 2 µL 10× RT Buffer, 0.44 µL MgCl_2_ 250 mM, 4 µL dNTP Mix, 0.4 µL Primer, 1 µL RNase inhibitor, 1 µL MultiScribe reverse transcriptase) and 10 µL of template (500 ng of RNA per sample and variable volume of H_2_O). The reaction was conducted with a T100™ Thermal Cycler (Bio-Rad, Copenhagen, Denmark) with the following program: 25 °C for 10 min, 37 °C for 60 min, 95 °C for 5 min, and 4 °C for ∞. Samples were diluted with 180 µL of DNase, RNase-free H_2_O. Samples were stored at −20 °C until gene expression analysis.

### 2.8. qPCR

#### 2.8.1. Fish Samples

Gene expression analysis of rainbow trout gill and spleen has been conducted on AriaMx Real-Time PCR Systems (Agilent Technologies, Glostrup, Denmark). A list of primers and probes used in this study is displayed in Table 2. The target gene of Ich *iag52a* was used to evaluate the parasite burden in fish gills. The final reaction volume for gene expression analysis was 12.5 µL and included 6.25 µL Brillant III Ultra-Fast QPCR Master Mix (Agilent Technologies, Glostrup, Denmark), 1 µL of primer mix (primer: 10 µM, probe: 5 µM), 2.75 µL of H_2_O, and 2.5 µL of cDNA template. The qPCR assays used in the present study exhibited efficiencies within 100 ± 5%, and data were analyzed using a simplified 2^−ΔΔCt^ method [13]. Thermal conditions consisted of a pre-denaturing step for 3 min at 95 °C followed by 40 cycles with denaturing for 5 s at 95 °C and elongation for 15 s at 60 °C.

#### 2.8.2. Water Sample

Water eDNA levels qPCR analysis of Ich was conducted with the gene *iag52a*, coding for an immobilization antigen. The eDNA analysis was done using the same machine as the fish analysis but using a different protocol. The final volume for analysis was 30 µL and included 15 µL TaqMan™ Environmental Master Mix 2.0 (Applied Biosystems, Lillerøde, Denmark ), 3 µL of primer mix (primer: 10 µM, probe: 5 µM), 7 µL of H_2_O, and 5 µL of DNA template. The thermal profile consisted of a pre-denaturing step for 10 min at 95 °C followed by 45 cycles with denaturing for 15 s at 95 °C and elongation for 60 s at 60 °C.

### 2.9. Statistics

#### 2.9.1. Fish Mortality

In this experiment, all statistical analyses were performed with the GraphPad Prism software (10.1.1). The fish mortalities were recorded in each duplicate infected tank. The statistical analysis was performed for infected groups only, as no fish died in the control groups. The log-rank (Mantel–Cox) test was used to calculate the statistical significance between each infected tank.

#### 2.9.2. Parasite Burden and Infection Severity

The trophont counts from skin and gill and the infection severity (Ich *iag52a* Ct value from rainbow trout gill) datasets were tested for normal distribution with the Shapiro–Wilk normality test. The infection severity was defined with the Ct value of the Ich target gene *iag52a* as molecular techniques such as qPCR are more reliable and complementary to macroscopical trophont count. For the trophont count, the datasets were not normal distributed and a non-parametric Mann–Whitney test was used, and for the *iag52a* Ct value from gill, the datasets were normal distributed and Student’s *t*-test was used to compare infected and infected-stress groups.

#### 2.9.3. Fish Gene Expression

For the gene expression analysis, the internal calibrator consisted of an average of two reference genes. Three reference genes (*arp*, *elf-1α*, and *β-actin*) were tested with NormFinder [23], and the most stable combination (i.e., *arp* and *elf-1α*) was used to perform calculations in this study. For each gene, fish organ, and fish treatments, outlier Ct values from qPCR analysis were removed using the GraphPad Outlier calculator (alpha = 0.05). The datasets were analyzed with the Shapiro–Wilk normality test to evaluate the normal (Gaussian) distribution. If the data were normally distributed, a one-way ANOVA with Tukey’s multiple comparisons test was conducted. Alternatively, a non-parametric Kruskal–Wallis test with multiple comparisons was performed. The evaluation of differential immune gene regulation induced by the unpredictable repeated stress protocol was conducted by comparing the ΔΔCt with either a Student’s *t*-test for parametric datasets or Mann-Whitney for non-parametric datasets. Only significant results (*p* < 0.05) were included in the Results section.

#### 2.9.4. Association between Parasite Burden, Infection Severity and Gene Expression

The relationship between spot count, immune response magnitude, and infection severity was determined with a correlation analysis. For each fish, both trophont (gill and skin) count and immune gene ΔΔCt (gill and spleen) were correlated with *iag52a* Ct value from fish gill. The datasets were tested for normal distribution with the Shapiro–Wilk normality test. Datasets following normal (i.e., Gaussian) distribution were analyzed with a parametric Pearson correlation test, and alternatively, a non-parametric Spearman correlation test was used. Only significant results (*p* < 0.05) were included in the Results section.

### 2.10. Ethical Statement

Experimental procedures were performed according to a license issued by the Experimental Animal Inspectorate, Ministry of Environment and Food, Denmark, with license number 2019-15-0201-00388.

## 3. Results

### 3.1. Fish Cumulative Mortality

The mortality in the infected fish treatment started at 10 dpi (Figure 1) and ranged from 64 to 80% between the two replicates. At 11 dpi, the mortality reached 100% in both replicates of the infected fish treatment. In the infected-stress fish treatment, significant differences were calculated (*p* < 0.0001) between the replicates. For the infected-stress 1 replicate, the mortality was 84% at 10 dpi and 100% at 11 dpi. For the infected-stress 2 replicate, the mortality was 52% at 11 dpi and 100% at 12 dpi. The infected-stress 2 replicate was significantly different (*p* < 0.0001) compared to both replicates of the infected fish treatment as mortalities were delayed.

### 3.2. Parasite Count, Burden and eDNA Levels

The trophont count in skin and gill (Figure 2A) showed that the gill was colonized more heavily than skin. In skin, an average of 3.80 (±4.16) trophonts were counted in the infected fish, whereas 1.00 (±2.21) were present in the infected-stress fish group. In gill, the average trophont count was 12.50 (±3.54) and 13.50 (±4.12) for infected and infected-stress fish, respectively. No statistical difference was detected between the two fish groups for both trophont counts. In gill, the Ct value of Ich *iag52a* was 33.36 (±1.24) in the infected fish and 33.15 (±1.00) in the infected-stress fish (Figure 2B). No statistically significant difference was detected between the two treatments. In the water samples, the Ct value was 26.08 (±0.49) and 26.57 (±0.86) for infected and infected-stress fish groups, respectively, at 7 dpi. The presence of Ich was not detected in the control groups with qPCR analysis and trophont count. 

### 3.3. Immune Gene Expression in Gill

The gill immune gene expression analysis indicated predominantly upregulation in both infected and infected-stress fish groups (Figure 3). The immune genes *il-4*/*13a*, *il-8*, *il-10*, *ifn-γ*, *hep*, *cat-1,* and *cat-2* were upregulated in both fish treatments compared to their respective control (Figure 3A,C,D,F,I–K). In the infected fish group, *il-6* was upregulated (Figure 3B), and in infected-stress fish groups, *tnf-α* was upregulated (Figure 3H) compared to their respective control. Differences in gene expression regulation between the two fish groups were reported for *il-4*/*13a*, *il-10*, *igt*, *ifn-γ*, *saa,* and *tnf-α* (Figure 3A,D–H). 

### 3.4. Immune Gene Expression in Spleen

The spleen gene expression analysis indicated predominantly downregulation of immune genes in the infected-stress fish group (Figure 4). In the infected fish group, *il-2* and *hep* were upregulated (Figure 4A,H) and *igt* was downregulated (Figure 4E). In the infected-stress fish group, *il-2*, *il-4*/*13a*, *il-8*, *ifn-γ*, *tnf-α*, *hep,* and *mhc-2* were downregulated (Figure 4A,B,D,F–I). Differences in gene expression regulation between the two fish groups were reported for *il-2*, *il-4*/*13a*, *il-6*, *hep,* and *mhc-2* (Figure 4A–C,H,I).

### 3.5. Association between Trophont Count, Immune Gene Expression and Infection Severity

Several immune genes were significantly positively correlated with infection severity. In the infected-stress fish group *c3*, *il-4*/*13a*, *il-8*, and *cat-2* were positively correlated (Figure 5B,E–G) and in infected fish group *igm* and *mhc-2* were positively correlated (Figure 5A,C,D) fish group. The immune genes correlated with infection severity were predominantly represented in spleen. Five immune genes were significantly positively correlated with infection severity in spleen and two immune genes in gill. In gill *igm* and *c3* were significantly positively correlated for respectively infected and infected-stress fish treatments. In spleen, *igm* and *mhc-2* were significantly correlated in infected treatment, and *il-4*/*13a*, *il-8,* and *cat-2* were significantly correlated in infected-stress treatment. There was no significant association correlation analysis between trophont count (gill and skin) and the infection severity (*iag52a* Ct in gill).

## 4. Discussion

In the present study, we evaluated the effects of unpredictable repeated stress on the rainbow trout immune response against the ciliate *Ichthyophthirius multifiliis*. The immune gene expression was evaluated in gill and spleen, and the pathogen burden was assessed using trophont counts in gill and skin and qPCR targeting *iag52a* from Ich in water and fish gill.

### 4.1. Rainbow Trout Typical Immune Gene Expression against Ichthyophthirius multifiliis

The local (i.e., gill) and systemic (i.e., spleen) immune responses against Ich differed. The only common upregulation for both gill and spleen was the cationic peptide *hep*. This peptide is known to play a role in iron regulation and antimicrobial activity [24]. This gene was upregulated in rainbow trout larvae (10 days post-hatching) exposed to Ich [25]. The role of *hep* in iron regulation may be predominant, as Ich infection in rainbow trout causes disturbance in osmoregulation, impairs wound healing ability of the skin, and induces anemia [26]. The two other antimicrobial peptides (AMPs) investigated in this study, *cat-1* and *cat-2*, were strongly upregulated in the gills of the infected fish, indicating that they play a role in the defense mechanisms against Ich infection. A study that investigated the rainbow trout immune response against Ich in different immune-related organs (ocular mucosa, head kidney, spleen) reported that *cat-1* was strongly upregulated in all sampled organs at 7 dpi [27]. These results are concordant with ours in gill at 7 dpi in both fish treatments. In addition, in rainbow trout larvae challenged with Ich, *cat-2* was one of the earliest genes upregulated at 12 h post-infection [25]. In our experiment, several markers of inflammation (pro and anti) regulation (*il-6*, *il-8*, *il-10*, *ifn-γ*) were upregulated in gill as described in previous studies. For example, pro-inflammatory cytokines, including *il-8* were upregulated in juvenile rainbow trout, indicating that Ich initiates an inflammatory reaction [28]. In rainbow trout larvae exposed to Ich, *il-6* and *il-8* were upregulated, and among them, *il-8* upregulation was one of the fastest to occur [25]. In a transcriptomic analysis of rainbow trout fry gills infected with Ich, 16 immune-related pathways were differentially regulated, and more precisely, elevated transcription of C-X-C chemokine ligand 8 (also known as *il-8*) indicated the rainbow trout initiated an inflammatory response after Ich exposure [29]. Another rainbow trout gills transcriptomic analysis revealed that *il-8* was included in the 25 most significantly upregulated genes, confirming the crucial role of *il-8* against Ich infection in rainbow trout gill [30].

### 4.2. Rainbow Trout Impaired Immune Gene Expression against Ichthyophthirius multifiliis Induced by Unpredictable Repeated Stress

In the present experiment, the effect of unpredictable repeated stress on immune gene expression in rainbow trout infected with Ich was differential in an organ-dependent manner between gill and spleen. In rainbow trout spleen, *il-4*/*13a* (molecular marker of Th2 response) and *il-8* downregulation were significantly positively correlated with the infection severity (Ct value in gill of Ich *iag52a*), indicating that the unpredictable repeated stress protocol impacted the rainbow trout ability to address a normal (i.e., infected fish group) immune response. The most noticeable difference was observed in the spleen, where *il-2* (a molecular marker of Th1 response) and *hep* were upregulated in the infected fish group and downregulated in the infected-stress fish group, showing an effect of unpredictable repeated stress in the context of Ich infection.

In previous studies, the effect of stress on fish immune gene expression was explored. An in vitro study demonstrated a downregulation of *hep* in liver slices of rainbow trout incubated with cortisol (100 ng/mL) [31]. Among all the stressors studied in the rainbow trout, the most explored one has been the effect of crowding. Thirty days of crowding (40 to 80 kg/m^3^) induced downregulation of immune genes (*tnf-1α*, *il-8*, *ifn-γ1,* and *lyzII*) in rainbow trout head kidney [32]. Nonetheless, the crowding effect on the modulation of rainbow trout immune gene expression appeared to be time dependent. In the case of long-term (from 240 to 300 days) and mild crowding (40 to 50 kg/m^3^), upregulation of *tnf-1α* and *il-10* in rainbow trout head kidney was reported [33]. Our findings indicated that in the spleen, *tnf-α*, *il-8*, and *ifn-γ* were downregulated, in agreement with the previously observed effects of short-term crowding. The effect of air exposure on *tnf-α* gene expression in rainbow trout showed upregulation in liver but not in spleen [34]. Our results reported a downregulation of *tnf-α* in the spleen. This finding could be due to the fish’s ability to cope with the stress, as the fish originated from hatchery and therefore have been exposed to handling. In rainbow trout gills, transportation (4 h, 50 fish per 30 L bag, aerated with pure oxygen) resulted in upregulation of *il-8* and *tnf-α* [35]. In our experiment, *il-8* was equally upregulated for both stressed and not stressed fish; however, *tnf-α* gene expression was upregulated only in the infected-stress fish.

### 4.3. Rainbow Trout Mortality and Parasite Burden in Relation with Ich Water eDNA Levels

Experimental challenges with Ich are difficult. To the best of our knowledge and despite several attempts, it is not possible to maintain an Ich isolate without the perpetual sacrifice of a fish host [36]. Nielsen and Buchmann (2000) demonstrated that the parasite life cycle could be partially completed in vitro with EPC (Epithelioma Papulosum Cyprini) cells and cell culture media (E-MEM or L-15); however, encystment of tomonts and tomite formation remain to be achieved [37].

Moreover, of the five serotypes documented for Ich (i.e., A, B, C, D, and E), some were reported to be more virulent than others, but published challenge studies rarely identify the isolate serotype [1,38,39]. Consequently, every experimental challenge could have uncertain outcomes, particularly for mortality and host immune response. Kong et al. (2023) recorded mortality of rainbow trout for 30 days, and at 14 dpi, mortality in the infected group (5000 theronts per fish; 12.5 g; 16 °C) ceased [27]. Buchmann et al. (2022) performed an Ich infection (186 theronts per fish; 13 °C) of rainbow trout (8 g), and all experimental groups died within four days (from 18 to 22 dpi) [40]. In the present study (500 theronts per fish; 12.88 g; 18 °C), both fish groups (infected and infected-stress) reached 100% mortality between 10 and 12 dpi, demonstrating that our isolate was particularly virulent. The eDNA results (Appendix A) demonstrated that the mortalities occurred during the second cycle of infection, which occurred early considering the temperature and the initial concentration of Ich theronts. Previous studies reported that a single tomont could result in up to 1000 theronts and that warmer water temperature resulted in larger tomonts and an increased number of theronts per tomonts (Ewing et al., 1986; Matthews, 2005) [1,41]. In addition, high water hardness (120 mg per L of CaCO_3_) improved Ich survival [3,42], and calcium was required for theronts but not tomonts [43]. In our facility, water hardness was around 500 mg per L of CaCO_3_, and this may have contributed to the high virulence of our Ich isolate. The calculation of theronts (Appendix A) in accordance with the established standard curve (Appendix A) showed that between 7 and 12 dpi, the theronts concentration in the tank water increased on average from 4290 theronts per L to 267,500 theronts per L, suggesting that the Ich isolate used in this study had an elevated infection rate. This result should be interpreted with caution, as eDNA could originate from dead tomonts that did not complete their development and theronts that failed to infect the fish.

Handling of fish has been associated with an increased susceptibility to Ich [44]. Nonetheless, our findings did not support this hypothesis. Despite no statistical difference in trophont count (gill and skin) and the Ct values of expression levels of *iag52a* in water eDNA and gill samples, it is important to note that the white spots were predominantly counted in rainbow trout gill compared to skin in both infected and infected-stress fish groups. Previous studies reported that the gill and skin of rainbow trout were damaged during the infection, resulting in a local inflammatory response [45]. Heavily infected gill results in failure of the epithelia, leading to respiratory and osmotic distress [46]. This outcome is similar in amoebic gill disease (AGD), where the presence of the *Neoparamoeba perurans* on Atlantic salmon (*Salmo salar*) gill induced inflammation and lamellar fusion, resulting in lower oxygen uptake and respiratory distress [47].

In conclusion, the unpredictable repeated stress protocol used in this study did not induce distress and an immunocompromised condition in rainbow trout that may have induced excess mortality. Our results indicated that stress caused by moderate (twice a day) aquaculture practices (e.g., air exposure, chasing, and transfer) would probably not immunocompromise rainbow trout.

## Figures and Tables

**Figure 1 biology-13-00769-f001:**
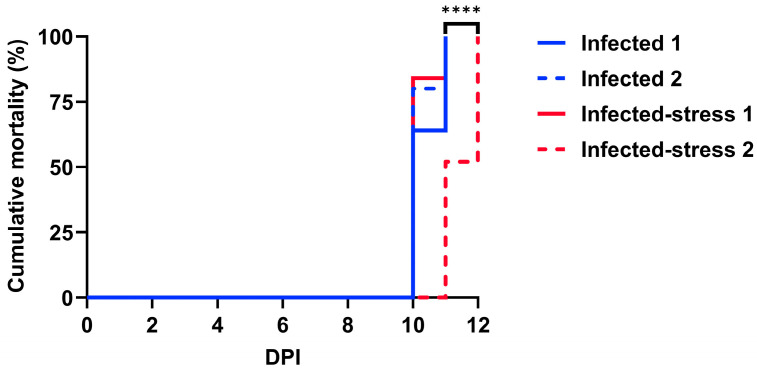
Recorded cumulative mortality in percentage of rainbow trout after exposure to Ich theronts (500 per fish). Each treatment was composed of 60 fish (*n* = 30 fish per tank). The tank duplicates of each fish treatment are represented separately dpi: days post infection. ****: *p* < 0.0001.

**Figure 2 biology-13-00769-f002:**
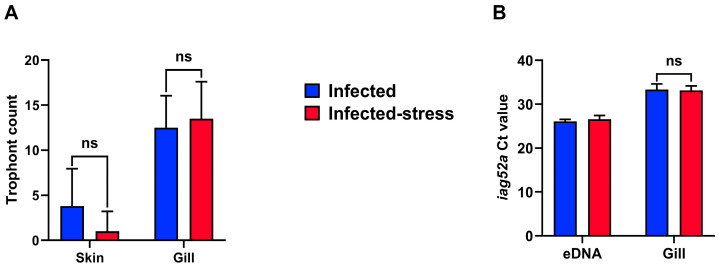
Evaluation of the Ich burden in fish (*n* = 5 fish per tank; 10 fish per treatment) and water at 7 dpi. (**A**) Ich spot count in rainbow trout skin and gill. (**B**) *iag52a* Ct value from water sampling (*n* = 1 per tank) and fish gill. ns: no statistical significance. No statistical analysis was performed for eDNA results due to limited sampling size.

**Figure 3 biology-13-00769-f003:**
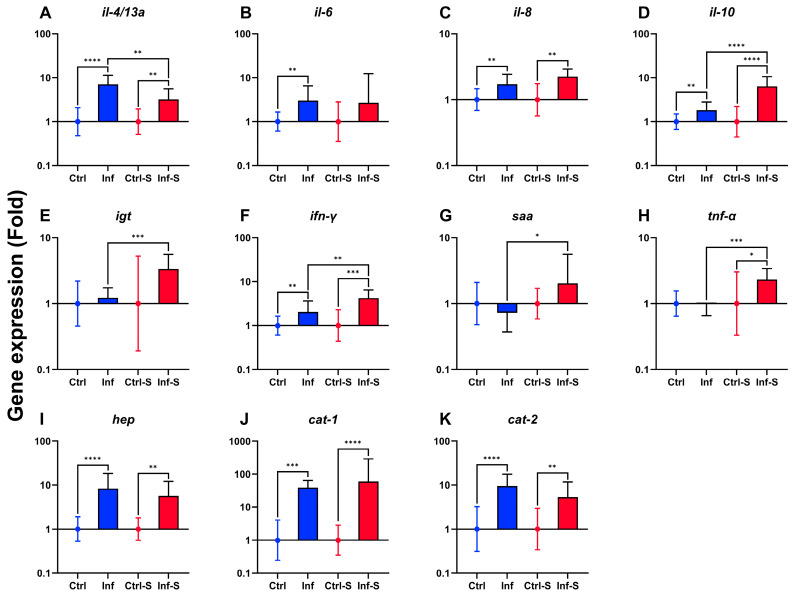
Gene expression of immune relevant gene (**A**–**K**) in rainbow trout gill at 7 dpi (*n* = 5 fish per tank; 10 fish per treatment). Data are represented as geometric mean with geometric standard deviation (*n* = 10 fish per group). Only significant results were represented. Ctrl: control. Inf: infected. S: stress. *: *p* < 0.05; **: *p* < 0.01; ***: *p* < 0.001; ****: *p* < 0.0001.

**Figure 4 biology-13-00769-f004:**
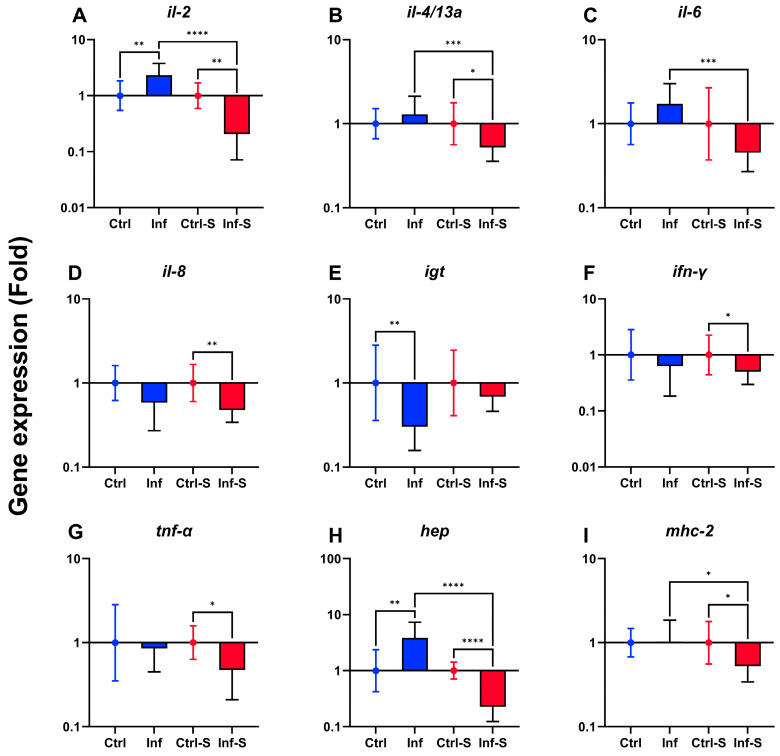
Gene expression of immune relevant gene (**A**–**I**) in rainbow trout spleen at 7 dpi (*n* = 5 fish per tank; 10 fish per treatment). Data are represented as geometric mean with geometric standard deviation (*n* = 10 fish per group). Only significant results were represented. Ctrl: control. Inf: infected. S: stress. *: *p* < 0.05; **: *p* < 0.01; ***: *p* < 0.001; ****: *p* < 0.0001.

**Figure 5 biology-13-00769-f005:**
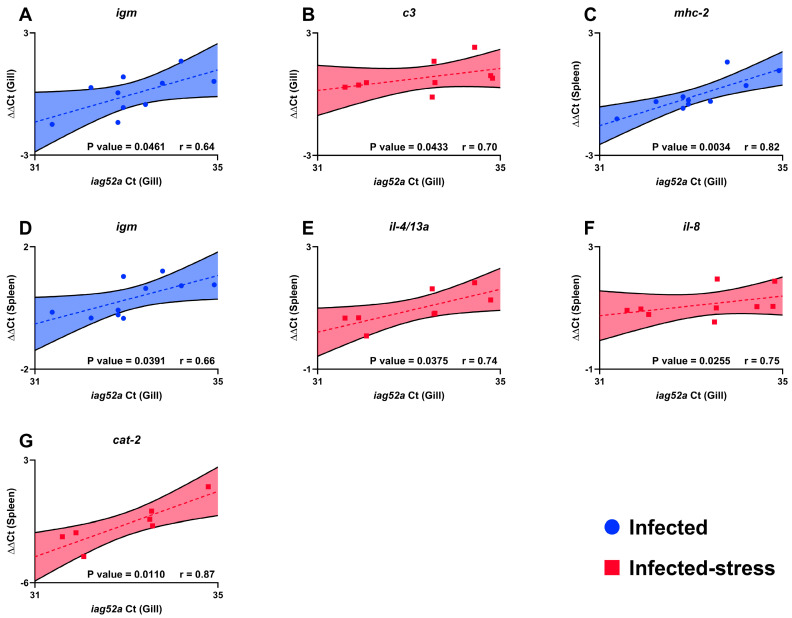
Correlation analysis of immune gene expression (gill and spleen) and infection severity (gill) measured with Ct value of *iag52a* (**A**–**G**) from Ich at 7 dpi (*n* = 5 fish per tank; 10 fish per treatment). Dots represent results from a single fish. Dotted lines represent simple linear regressions and colored areas represent 95% confidence intervals associated with each regression. Only significant results (*p* < 0.05) were presented.

**Table 1 biology-13-00769-t001:** Unpredictable repeated stress protocol performed over the course of the experiment in control-stress and infected-stress fish groups. Air exposure: the fish were placed in a net (295 × 295 × 250 mm) and suspended in air for 20 s. Chasing: the fish were chased with a net for 2 min. Transfer: the fish were transferred in a 20 L tank for 2 min then moved back to their respective tank.

Day Post Infection		Air Exposure	Chasing	Transfer
−3		3:15 p.m.		
−2			3:40 p.m.	
−1				2:25 p.m.
0	Infection			
1		10:10 a.m.	1:10 p.m.	
2		3:30 p.m.		1:35 p.m.
3			10:30 a.m.	3:15 p.m.
4		9:55 a.m.	3:15 p.m.	
5		1:35 p.m.		10:45 a.m.
6			9:25 a.m.	3:35 p.m.
7	Sampling			
8		10:22 a.m.	1:55 p.m.	
9				1:40 p.m.
10			2:40 p.m.	
11				
12	Experiment end			

**Table 2 biology-13-00769-t002:** Primers used for eDNA and gene expression analysis. F: forward. R: reverse. P: probe. *elf*: elongation factor. *arp*: acidic ribosomal phosphoprotein. *il*: interleukin. *ig*: immunoglobulin. *ifn*: interferon. *tnf*: tumor necrosis factor. *mhc*: major histocompatibility complex. *hep*: hepcidin. *cat*: cathelicidin. *iag*: immobilization antigen.

Gene	Species	Sequence 5′-3′	Product Size (bp)	Ref.
GenBank Acc. Num				
*β-actin*	*O. mykiss*	F: ACATCAAGGAGAAGCTGTGCTAC	241	[14]
AB196465		R: TACGGATGTCCACGTCACAC		
		P: CCTCTCTGGAGAAGAGCTACGAGCTG		
*elf-1*α	*O. mykiss*	F: ACCCTCCTCTTGGTCGTTTC	63	[15]
AF498320		R: TGATGACACCAACAGCAACA		
		P: GCTGTGCGTGACATGAGGCA		
*arp*	*O. mykiss*	F: GAAAATCATCCAATTGCTGGATG	106	[16]
AY505012		R: CTTCCCACGCAAGGACAGA		
		P: CTATCCCAAATGTTTCATTGTCGGCGC		
*il-1*β	*O. mykiss*	F: ACATTGCCAACCTCATCATCG	91	[17]
AJ223954		R: TTGAGCAGGTCCTTGTCCTTG		
		P: CATGGAGAGGTTAAAGGGTGGC		
*il-2*	*O. mykiss*	F: ATGCAACACCACATCAGCAT	110	[14]
FJ571513		R: TTTTCCAACCGCTTGTCTTC		
		P: TGCCACGGCCCTACAAAAGA		
*il-4*/*13a*	*O. mykiss*	F: ATCCTTCTCCTCTCTGTTGC	139	[18]
AB574337		R: GAGTGTGTGTGTATTGTCCTG		
		P: CGCACCGGCAGCATAGAAGT		
*il-6*	*O. mykiss*	F: ACTCCCCTCTGTCACACACC	91	[17]
DQ866150		R: GGCAGACAGGTCCTCCACTA		
		P: CCACTGTGCTGATAGGGCTGG		
*il-8*	*O. mykiss*	F: AGAATGTCAGCCAGCCTTGT	69	[17]
AJ279069		R: TCTCAGACTCATCCCCTCAGT		
		P: TTGTGCTCCTGGCCCTCCTGA		
*il-10*	*O. mykiss*	F: CGACTTTAAATCTCCCATCGAC	70	[15]
AB118099		R: GCATTGGACGATCTCTTTCTTC		
		P: CATCGGAAACATCTTCCACGAGCT		
*il-22*	*O. mykiss*	F: ATGACCACCACCACAGCATT	64	[19]
AM748537		R: ATTCCTTTCCCCTCCTCCAT		
		P: CTTTCCGCAAGAAGTTGTCCGAG		
*igm*	*O. mykiss*	F: CTTGGCTTGTTGACGATGAG	72	[15]
AH014877—S63348		R: GGCTAGTGGTGTTGAATTGG		
		P: TGGAGAGAACGAGCAGTTCAGCA		
*igt*	*O. mykiss*	F: AGCACCAGGGTGAAACCA	73	[15]
AY870265—AY870263		R: GCGGTGGGTTCAGAGTCA		
		P: AGCAAGACGACCTCCAAAACAGAAC		
*ifn*-γ	*O. mykiss*	F: AAGGGCTGTGATGTGTTTCTG	68	[17]
FJ184374/FJ184375		R: TGTACTGAGCGGCATTACTCC		
		P: TTGATGGGCTGGATGACTTTAGGA		
*saa*	*O. mykiss*	F: GGGAGATGATTCAGGGTTCCA	79	[20]
AM422446		R: TTACGTCCCCAGTGGTTAGC		
		P: TCGAGGACACGAGGACTCAGCA		
*tnf*-α	*O. mykiss*	F: GGGGACAAACTGTGGACTGA	75	[17]
AJ277604/AJ401377		R: GAAGTTCTTGCCCTGCTCTG		
		P: GACCAATCGACTGACCGACGTGGA		
*c3*	*O. mykiss*	F: ATTGGCCTGTCCAAAACACA	85	[20]
AF271080		R: AGCTTCAGATCAAGGAAGAAGTTC		
		P: TGGAATCTGTGTGTCTGAACCCC		
*mhc-2*	*O. mykiss*	F: TGCCATGCTGATGTGCAG	68	[15]
AF115533		R: GTCCCTCAGCCAGGTCACT		
		P: CGCCTATGACTTCTACCCCAAACAAAT		
*hep*	*O. mykiss*	F: GAGGAGGTTGGAAGCATTGA	95	[20]
AF281354		R: TGACGCTTGAACCTGAAATG		
		P: AGTCCAGTTGGGGAACATCAACAG		
*cox-2*	*O. mykiss*	F: GGGCTTTGACATCCTCAACA	73	[15]
NM_001124348.1		R: CATCGGACAAGAACCCTTGA		
		P: CTTCATTGGAGAGGCTGGTGTGC		
*cat-1*	*O. mykiss*	F: TCTCTCGTCCTGGGGTT	189	[14]
AY382478		R: GTTGTAGCGTGCTGATCTATG		
		P: TAATTGGTCGTCCTGGGGGTGG		
*cat-2*	*O. mykiss*	F: AAAGATTCCAAGGGGGGT	135	[21]
AY360356		R: CAAAGGGTGTGTTGTGCTGT		
		P: GCTCTCGTCCTGGGTTTGGCTCC		
*iag52a*	*I. mutifiliis*	F: TTGGAACTGAAACTAACACAGCC	240	[22]
AF324424		R: CTCCACCTGCAATTGCGGTA		
		P: TGCTGCTGCTTTCGTTCCTGGTGC		

## Data Availability

Data will be made available upon request.

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
