# Peer review of "Unpredictable Repeated Stress in Rainbow Trout (*Oncorhynchus mykiss*) Shifted the Immune Response against a Fish Parasite"

_biology, 2024, doi:10.3390/biology13100769_

Round 1
Reviewer 1 Report
Comments and Suggestions for Authors
The paper/study is relatively straightforward. I only have a few comments. The most substantive one is about the statistics - I'm not super sure the tests used were the most appropriate to address the dependence of fish within tanks and also for the analysis of the count data. I mention this again in my specific comments but generalized linear and generalized linear mixed models might be required for some of your data.
Line 2: not a great start for me to set the tone for the rest of my comments, but I’m not sure ‘unpredictable’ is the best word here. It suggests there is some prediction going on, but it’s not clear who is doing it: you or the fish. Something like ‘stochastic’, ‘random timing of’, or something else might get closer to your design.
Line 11: ‘represent’ can be replaced with ‘is’
Line 15: presumably you used several treatments/types of handling? Might be worth mentioning how many treatments (and comparison to a ‘no handling’ scenario)
Lines 23-24: I’m confused by this description – this seems to suggest all fish were subjected to all handling procedures, just in a different order?
Line 28: two groups? You said three on line 24.
Lines 105: again, I’m not sure how to best describe this procedure. It’s more pseudo-random as described here – yes, there are two time frames, but there are ‘stress windows’ so the ‘unpredictable’ part of the stress timing is constrained. I want to be clear. I’m not discounting what you did, I’m just not sure your description is ideal. It’s also worth mentioning though (and maybe you did and I missed it) but why these timing windows? Are these feeding times in fish farms?
Lines 186-223: something to watch for is the lack of independence between your fish within tanks, I’m not totally plugged into all of the stats about this problem, but certainly hierarchical/mixed models might be a better choice for some of these analyses.
Lines 198-201: using a statistical distribution appropriate for count data is probably better (such as the Poisson in a generalized linear model – and probably a generalized linear mixed model with fish nested within tank)
Line 233: should this be p<0.0001? Same with line 236. p>0.0001 could still be statistically significant at alpha=0.05.
Lines 247-248: on line 223, you say you only mention significant results; I suggest deleting line 223.
Figure 5: since you say ‘correlation analysis’ I suggest adding an r value to this – spearman’s rank might be the best choice, but that’s up to you.
Lines 338-353: this is mostly results and should either be moved, removed, or substantially reduced for the discussion
Lines 354-370: This paragraph is missing a topic sentence. How does this relate to your work?
Line 393: you can probably start a new paragraph here
Line 413: before you get to the conclusion, I think it’s worth bringing your results back to the fish farms you mentioned in the introduction – importantly, I think its worth discussing how applicable your results are to the handling conditions at a fish farm. For example, your results suggest handling stress is not a major driver of Ich infection in fish farms, but your study also did not replicate a fish farm. This sort of context will help your paper.
Comments on the Quality of English Language
english is fine
Author Response
Comment 1: Line 2: not a great start for me to set the tone for the rest of my comments, but I’m not sure ‘unpredictable’ is the best word here. It suggests there is some prediction going on, but it’s not clear who is doing it: you or the fish. Something like ‘stochastic’, ‘random timing of’, or something else might get closer to your design.
Response 1: We understand your point however, we believe that unpredictable is the most appropriate term. Our study is inspired by the work of Piato et al., 2011 (doi:10.1016/j.pnpbp.2010.12.018) and the title of that study was: “Unpredictable chronic stress model in zebrafish (Danio rerio): Behavioral and physiological responses”. Consequently, as the stress protocol we use is adapted from that study, we decided to conserve the wording.
Comment 2: Line 11: ‘represent’ can be replaced with ‘is’
Response 2: In L.11 “represent” was replaced by “is” according to reviewer recommendation: “…caused by Ichthyophthirius multifiliis (ich) is a significant threat to fish production and welfare…”
Comment 3: Line 15: presumably you used several treatments/types of handling? Might be worth mentioning how many treatments (and comparison to a ‘no handling’ scenario)
Response 3: L.15 was updated. The three procedures were indicated. We also mentioned that handled fish were compared to non-handled fish : “…the impact of handling procedures (air exposure, chasing, transfer) on the fish’s ability to fight the parasite compared to non-handled fish…”
Comment 4: Lines 23-24: I’m confused by this description – this seems to suggest all fish were subjected to all handling procedures, just in a different order?
Response 4: Indeed, this is what was performed in the study and described in the material and method. There were two conditions, handled and non-handled. The handled fish were handled with either air exposure, chasing and transfer twice a day as indicated in Figure 1.
Comment 5: Line 28: two groups? You said three on line 24.
Response 5: In L.28 we specified two groups because our experiments included two groups which were infected handled and infected non-handled fish as described in material and method section : “The four experimental treatments were performed in duplicate (n=30 fish/tank; n=60 fish /treatment) tanks and defined as: control, control-stress, infected, infected-stress.”
Comment 6: Lines 105: again, I’m not sure how to best describe this procedure. It’s more pseudo-random as described here – yes, there are two time frames, but there are ‘stress windows’ so the ‘unpredictable’ part of the stress timing is constrained. I want to be clear. I’m not discounting what you did, I’m just not sure your description is ideal. It’s also worth mentioning though (and maybe you did and I missed it) but why these timing windows? Are these feeding times in fish farms?
Response 6: We get your point regarding the pseudo-random. It is true that only the time frames were strictly randomised. However, the procedures were repeated according to a pattern that takes 3 days to be completed. The handled fish experienced the full pattern only once and the procedures were always different in the morning and the afternoon. Therefore, the fish did not get time to adapt and predict the procedures. The timing was selected to make sure that the fish were back to normal after the peak of cortisol induced by the handling procedures. The material and method was updated to mention this point : “…The two time frames were selected to provide enough time for the fish to recover from the handling procedure…”
Comment 7: Lines 186-223: something to watch for is the lack of independence between your fish within tanks, I’m not totally plugged into all of the stats about this problem, but certainly hierarchical/mixed models might be a better choice for some of these analyses.
Response 7: The sampling procedure is standardized in the field of fish infection studies. Similar setup could be found in published studies: Syahputra et al., 2019 (https://doi.org/10.1016/j.fsi.2018.11.075), Yang et al., 2021 (DOI: 10.1111/jfd.13507).
Comment 8: Lines 198-201: using a statistical distribution appropriate for count data is probably better (such as the Poisson in a generalized linear model – and probably a generalized linear mixed model with fish nested within tank)
Response 8: To perform the statistical analysis, we based the method according to similar studies for spot count (Mathiessen et al., 2023 (https://doi.org/10.3389/fcimb.2023.1190931), gene expression analysis (Marana et al., 2021 (https://doi.org/10.1038/s41598-021-97437-7)) and the correlation analysis (Mérou et al., 2020 (doi: 10.1111/1751-7915.13617)). In addition the software employed to perform the statistical calculation (GraphPad Prism) provides guidelines to choose the right test according to the type of data and the characteristic of the setup. Therefore we investigated extensively how to process our data and we are confident about the choice we made that are supported by literature and software guidelines.
Comment 9: Line 233: should this be p<0.0001? Same with line 236. p>0.0001 could still be statistically significant at alpha=0.05.
Response 9: Thank you for noticing the typing mistake. The symbol “>” was replaced by “<” in the two mentioned lines : “…significant differences were calculated (p<0.0001) between…”; “…significantly different (p<0.0001) compared to both replicate…”
Comment 10: Lines 247-248: on line 223, you say you only mention significant results; I suggest deleting line 223.
Response 10: The mention “Only significant results (p<0.05) were included in results section” was indicated twice for the Fish gene expression analysis (L. 219) and the correlation analysis (L. 228) because some immune gene and correlation were not significant. We decided to not include them in the results section to make the paper lighter. To clarify that the results were not forgotten, we decided to add this justification wherever applicable.
Comment 11: Figure 5: since you say ‘correlation analysis’ I suggest adding an r value to this – spearman’s rank might be the best choice, but that’s up to you.
Response 11: This is a very good point. The r value was added for every single significant correlation in Figure 5.
Comment 12: Lines 338-353: this is mostly results and should either be moved, removed, or substantially reduced for the discussion
Response 12: The paragraph was substantially reduced according to the reviewer suggestion: “In the present experiment, the effect of unpredictable repeated stress on immune gene expression in rainbow trout infected with Ich was differential in an organ-dependent manner between gill and spleen. In rainbow trout spleen, il-4/13a (molecular marker of Th2 response) and il-8 downregulation were significantly positively correlated with the infection severity (Ct value in gill of Ich iag52a) indicating that the unpredictable repeated stress protocol impacted the rainbow trout ability to address a normal (i.e. infected fish group) immune response. The most noticeable difference was observed in spleen where il-2 (molecular marker of Th1 response) and hep were upregulated in infected fish group and downregulated in infected-stress fish group, showing an effect of unpredictable repeated stress in the context of Ich infection.
In previous studies, the effect of stress on fish immune gene expression was explored. An in vitro study demonstrated a downregulation of hep in liver slices of rainbow trout incubated with cortisol (100 ng/mL) [31].”.
Comment 13: Lines 354-370: This paragraph is missing a topic sentence. How does this relate to your work?
Response 13: A topic sentence was inserted at the beginning of the paragraph: “…In previous studies, the effect of stress on fish immune gene expression was explored. An in vitro study demonstrated a downregulation…”
Comment 14Line 393: you can probably start a new paragraph here
Response 14: A new paragraph was started at the mentioned line: “This result should be interpreted with caution as eDNA could originate from dead tomonts which did not complete their development and theronts which failed to infect the fish.
Handling of fish has been associated with an increased susceptibility to Ich [44]. Nonetheless, our findings did not support this hypothesis.”
Comment 15: Line 413: before you get to the conclusion, I think it’s worth bringing your results back to the fish farms you mentioned in the introduction – importantly, I think its worth discussing how applicable your results are to the handling conditions at a fish farm. For example, your results suggest handling stress is not a major driver of Ich infection in fish farms, but your study also did not replicate a fish farm. This sort of context will help your paper.
Response 15: The main question that we attempted to answer to the best of our abilities in the study is, would the stress experience by the fish in a fish farm immunocompromise the fish and induces excess mortalities in a context of an outbreak. It is true that we could not replicate an authentic fish farm in our facilities however, the way the infection was performed and how the handling procedures were selected was fitting real farming conditions.
Reviewer 2 Report
Comments and Suggestions for Authors
Dear Authors,
The manuscript “Unpredictable Repeated Stress in Rainbow Trout Oncorhynchus mykiss Shifted the Immune Response against a Fish Parasite” addresses a relevant topic in the field. Stress plays a crucial role in the immune system.
The study is well-conducted, supported by relevant references, and includes an appropriate methodological description. The results obtained are directly linked to the discussion.
However, I have a few observations for the authors:
Methodology:
-
In the description of the animal treatment, there is a lack of detail regarding the replicates/duplicates shown in the results (Figure 1). It is unclear how this was conducted.
-
In the description of the sample collection for gene expression analysis, the authors should specify the number of animals per treatment used for RNA extraction. Additionally, clarify when this material was collected.
Results:
-
In Figure 1, cumulative mortality is presented, and the authors show data from the duplicates separately. Please explain the rationale for analyzing the duplicates separately instead of as a result of only two treatments (infected, infected and stressed).
Other Observations:
-
Consider modifying the colors used in the figures to enhance contrast. In Figure 5, use different symbols to distinguish between the two treatments.
Author Response
Comment 1: In the description of the animal treatment, there is a lack of detail regarding the replicates/duplicates shown in the results (Figure 1). It is unclear how this was conducted.
Response 1: The text in L. 94-95, and the legend of Figure 1 was updated to be clearer about the number of fish in tank and in each treatment: “…The four experimental treatments were performed in duplicate (n=30 fish/tank; n=60 fish /treatment) tanks…”
Comment 2: In the description of the sample collection for gene expression analysis, the authors should specify the number of animals per treatment used for RNA extraction. Additionally, clarify when this material was collected.
Response 2: In L.109, we specify that the sampling was performed at 7 dpi and that 5 fish were sampled per tank. From L.109-114, we describe the sampling procedure which included fish euthanasia, spot count in skin, spot count in gill, gill sampling, spleen sampling. Nonetheless, the legend of Figure 2; Figure 3; Figure 4 and Figure 5 were updated to improve clarity.
Comment 3: In Figure 1, cumulative mortality is presented, and the authors show data from the duplicates separately. Please explain the rationale for analyzing the duplicates separately instead of as a result of only two treatments (infected, infected and stressed).
Response 3: Thank you for rising this point as that was also a discussion between authors of the study. Before doing statistical calculation between fish treatments, the analysis is performed within the two duplicates. This is valid for gene expression, spot count and correlation analysis. We found significant differences between the two replicates of infected-stress and therefore, we decided to show the results from every single duplicates. If the results were gathered between replicates, we would have found a significant differences between infected and infected-stress groups and that would be misleading.
Comment 4: Consider modifying the colors used in the figures to enhance contrast. In Figure 5, use different symbols to distinguish between the two treatments.
Response 4: The colours were changed to increase the contrast and between infected and infected-stress groups in all the figures. In Figure 5, infected-stress symbol was changed with square instead of dot.
Reviewer 3 Report
Comments and Suggestions for Authors
In this manuscript, the authors investigated whether handling stress affects the rainbow trout immunity against Ich. The results showed that handling stress did not change the number of Ich or the fish mortality after Ich challenge. However, handling stress did alter the expression of several immune genes in Ich-infected fish; especially in the spleen, down-regulation of several immune genes was noted. I have only some minor suggestions for this manuscript, as indicated below.
1. In the Abstract, Lines 29-31: please rewrite the sentence as “In gills, a T helper cell type 2 immune response was initiated, whereas in spleen, a T helper cell type 1 immune response was observed.” And I did not find any related information presented in the Results or Discussion sections.
2. line 65: delete “,” after stress.
3. Lines 336-337: please change to “...stress on immune gene expression in rainbow trout infected with Ich was…”.
4. line 337: please change to “organ-dependent”.
5. line 339: please change to “infected-stress fish, whereas il-10…”.
6. Line 364: please change to “upregulation in liver”.
7. Line 374: please delete “,” after Buchmann (2000).
8. Line 382: please delete “,” after Kong et al. (2023).
9. Line 384: please delete “,” after Buchmann et al.
10. Line 413: please add “,” after In conclusion.
Comments on the Quality of English Languagesome puctuation problems.
Author Response
Comment 1. In the Abstract, Lines 29-31: please rewrite the sentence as “In gills, a T helper cell type 2 immune response was initiated, whereas in spleen, a T helper cell type 1 immune response was observed.” And I did not find any related information presented in the Results or Discussion sections.
Response 1: The sentence was rewrite according to the reviewer suggestion. The introduction was updated to indicate which molecular markers were selected to examine the Th1 and Th2 responses: “…Specific molecular markers involved in inflammation (e.g ifn-γ, il-1β, il-10), antimicrobial activity (e.g hep, cat-1, c3), Th1 (e.g il-2) and Th2 (e.g il-4/13a) responses were examined…”
The discussion was also updated L. to clarify marker of Th1 and Th2 response: “…In rainbow trout spleen, il-4/13a (molecular marker of Th2 response) and…”; “…in spleen where il-2 (molecular marker of Th1 response) and…”
Comment 2. Line 65: delete “,” after stress.
Response 2: The comma L. 65 was removed: “…response whereas acute stress boosts innate immune response…”
Comment 3. Lines 336-337: please change to “...stress on immune gene expression in rainbow trout infected with Ich was…”.
Response 3: The sentence was rewrite according to the reviewer suggestion: “…stress on immune gene expression in rainbow trout infected with Ich…”
Comment 4. Line 337: please change to “organ-dependent”.
Response 4: Organ dependent has been changed to organ-dependent: “…with Ich was differential in an organ-dependent manner between gill and spleen…”
Comment 5. Line 339: please change to “infected-stress fish, whereas il-10…”.
Response 5: The sentence was removed according to another reviewer comment.
Comment 6. Line 364: please change to “upregulation in liver”.
Response 6: Thank you for remarking that typing error between upregulation and in: “…rainbow trout showed upregulation in liver but not in spleen…”
Comment 7. Line 374: please delete “,” after Buchmann (2000).
Response 7: The comma was removed according to the reviewer suggestion: “…Nielsen and Buchmann (2000) demonstrated that the…”
Comment 8. Line 382: please delete “,” after Kong et al. (2023).
Response 8: The comma was removed according to the reviewer suggestion: “…Kong et al. (2023) recorded mortality of rainbow trout for 30 days…”
Comment 9. Line 384: please delete “,” after Buchmann et al.
Response 9: The comma was removed according to the reviewer suggestion: “…Buchmann et al. (2022) performed an Ich infection…”
Comment 10. Line 413: please add “,” after In conclusion.
Response 10: The comma was added according to the reviewer suggestion: “…In conclusion, the unpredictable repeated stress protocol…”
Round 2
Reviewer 1 Report
Comments and Suggestions for Authors
Thank you for addressing the comments and responding. Just a word of caution for future reference. Because other people have used particular stats or particular procedures, or a statistical software package makes particular recommendations, does not mean any are correct nor the most appropriate.
The issue I was getting at was the debate about psuedo-replication (e.g., dependence between observations). The debate tends to swing back and forth about pseudo-replication and how serious it is - in your case, pseudo-replication might not be such a problem since its effect is to over-power your analysis...your false positive rate was higher than 5%; in other words, you were more likely to see a statistically significant result because the fish were not true replicates. But even with this, you still saw no impact of the handling stress protocol. However, dependence among the observations (or sets of observations) would be more of an issue if you did see an effect between treatments.